# 3D Geometric Morphometrics Reveals Convergent Character Displacement in the Central European Contact Zone between Two Species of Hedgehogs (Genus *Erinaceus*)

**DOI:** 10.3390/ani10101803

**Published:** 2020-10-04

**Authors:** Barbora Černá Bolfíková, Allowen Evin, Markéta Rozkošná Knitlová, Miroslava Loudová, Anna Sztencel-Jabłonka, Wiesław Bogdanowicz, Pavel Hulva

**Affiliations:** 1Faculty of Tropical AgriSciences, Czech University of Life Sciences Prague, Kamýcká 129, 165 21 Prague, Czech Republic; 2Institut des Sciences de l’Evolution—Montpellier (ISEM), Univ Montpellier, CNRS, EPHE, IRD, 2 place Eugène Bataillon, CC065, CEDEX 5, 34095 Montpellier, France; allowen.evin@umontpellier.fr; 3Department of Zoology, Faculty of Science, Charles University in Prague, Viničná 7, 128 00 Prague, Czech Republic; knitlova@natur.cuni.cz (M.R.K.); miroslava.loudova@natur.cuni.cz (M.L.); hulva@natur.cuni.cz (P.H.); 4Museum and Institute of Zoology, Polish Academy of Sciences, Wilcza 64, 00-679 Warszawa, Poland; brysio@miiz.waw.pl (A.S.-J.); wieslawb@miiz.waw.pl (W.B.); 5Faculty of Science, University of Ostrava, Chittussiho 10, 710 00 Ostrava, Czech Republic

**Keywords:** convergent character displacement, *Erinaceus*, geometric morphometrics, species interactions

## Abstract

**Simple Summary:**

Hedgehogs, being insectivores with slow metabolisms, are quite sensitive to temperature and food availability. As a consequence, their ranges have oscillated in relation to past climate changes. Species that have evolved in different regions, but their ranges have shifted and overlapped subsequently, often represent intense competitors as a result of ecological similarities. The present study focuses on this phenomenon in the contact zone in central Europe and adjacent regions, using genetic determination of species and description of size and shape of skull, the morphological structure mirroring many selection pressures related to ecology. While animals living outside of the contact zone show marked differences between the two species, individuals within the contact zone are more alike with a smaller skull size and a convergent jawbone shape. Changes in skull size can be related to inter-species competition and also facilitated by selection pressure, mediated by overpopulated medium-sized predators such as foxes or badgers. Since the function of the lower jaw is mainly connected to feeding, we hypothesize that this pattern is due to the selection to size and shape related to competition for food resources. The present study helps to describe general patterns related to species formation, as well as species responses to anthropogenic environmental changes.

**Abstract:**

Hedgehogs, as medium-sized plantigrade insectivores with low basal metabolic rates and related defensive anti-predator strategies, are quite sensitive to temperature and ecosystem productivity. Their ranges therefore changed dramatically due to Pleistocene climate oscillations, resulting in allopatric speciation and the subsequent formation of secondary contact zones. Such interactions between closely related species are known to generate strong evolutionary forces responsible for niche differentiation. In this connection, here, we detail the results of research on the phenotypic evolution in the two species of hedgehog present in central Europe, as based on genetics and geometric morphometrics in samples along a longitudinal transect that includes the contact zone between the species. While in allopatry, *Erinaceus europaeus* is found to have a larger skull than *E. roumanicus* and distinct cranial and mandibular shapes; the members of the two species in sympatry are smaller and more similar to each other, with a convergent shape of the mandible. The relevant data fail to reveal any major role for either hybridisation or clinal variation. We, therefore, hypothesise that competitive pressure exerted on the studied species does not generate divergent selection sufficient for divergent character displacement to evolve, instead giving rise to convergent selection in the face of resource limitation in the direction of smaller skull size. Considering the multi-factorial constraints present in the relevant adaptive landscape, reduction in size could also be facilitated by predator pressure in ecosystems characterised by mesopredator release and other anthropogenic factors. As the function of the animals’ lower jaw is mainly connected with feeding (in contrast to the cranium whose functions are obviously more complex), we interpret the similarity in shape as reflecting local adaptations to overlapping dietary resources in the two species and hence as convergent character displacement.

## 1. Introduction

Hedgehogs from the Western Palearctic play a key role as model organisms in the field of phylogeography and speciation studies (e.g., [1,2,3]). As medium-sized insectivores, they are sensitive to temperature and thus, to the productivity of ecosystems, to the extent that their ranges changed substantially during the Pleistocene climatic oscillations [2]. Indeed, isolation in southern refugia facilitated allopatric speciation scenarios, resulting in a recent pattern of east–west parapatry, with the northern white-breasted hedgehog (*Erinaceus roumanicus*) present in eastern Europe, and the European hedgehog (*E. europaeus*) occurring in western Europe. The two species form a secondary contact zone in central Europe (Italy, Austria, Czech Republic and Poland), with a relatively broad area of sympatry in the center of the zone (Czech Republic), possibly in relation to Neolithic deforestation [3]. This region therefore provides conditions suitable for studying species interactions in relation to genomic and ecological niche-differentiation in the context of anthropogenic environmental changes.

There has been little gene flow between *E. europaeus* and *E. roumanicus* [3,4,5]. Alongside genetic differentiation, there is a divergence of phenotypic traits in the closely related species which is an important source of information about adaptive processes, i.a. indicating nascent niche diversification [6]. In general, phenotypes respond to both abiotic and biotic factors. In sympatry, environmental factors are identical, but ecological, microallopatric or trophic differentiation may occur [7]. The integration of originating species into ecological networks also varies during the speciation process, in line with an increasing role for competition with the sister species that may facilitate the niche differentiation [8].

However, phenotypic variation in species with extensive ranges usually shows pronounced geographical variation, which complicates the comparison of allopatric populations. For example, the body size of *E. roumanicus* increases linearly from north to south in Europe and is thus shown to correlate positively with temperature, as well as negatively with precipitation in the summer [9]. It is hypothesised that size is in this case determined by seasonality of the resource availability [9].

Comparisons between the two species in allopatry thus tend to be lacking, given the requirement of sampling across the entire range, and/or controlled design of the study and homologous comparison, as regards environmental impact. Reeve [10] states that the species are of approximately the same size and weight, while comparisons between populations from Great Britain, Italy, Switzerland, Germany, Poland, Russia and Ukraine by Ruprecht [11] failed to find a key character allowing the species to be distinguished. The above author only sums up that “the western species is more differentiated in skull dimensions than the eastern species”. Phenotypic differences between the white-breasted and European hedgehogs by reference to traditional morphometrics, or non-metric, discrete characters in the cranial phenotype prove relatively difficult. Hedgehogs are traditionally distinguished by cranial indices and the length of naso-maxillary sutures [12,13,14]; these characters in fact show marked intraspecific variability and interspecific overlap. Equally, sexual dimorphism related to size has never been found in either *E. roumanicus* or *E. europaeus* [15,16].

In sympatry, *E. europaeus* has a higher mean body mass, mean body length, mean hindfoot length and mean ear length than *E. roumanicus* in all adult age classes, as well as higher values for cranial indexes [15,17,18]. *Erinaceus roumanicus* has a longer tail in adult categories [17]. Body mass and neurocranial capacity in turn increase at a higher rate in *E. roumanicus* [17,19]. Deciduous dentition is replaced earlier in *E. roumanicus* than in *E. europaeus*, while also showing fewer deviations from the normal dental formula [20].

The objective of this research was to compare phenotypic differences in allopatric and sympatric populations of the above-mentioned species in terms of size and shape of their skulls, by reference to geometric morphometrics and genetic determination of species. The skull, as a complex morphological structure integrating traits associated with cognitive, sensory and food-processing functions, reflects the diverse selection pressures associated with the ecological niche [6,21]. In order to eliminate the effect of environmental adaptation (position in relation to the north–south and oceanic–continental gradients), we used longitudinal transect sampling spatially crossing the relatively bounded region in central Europe. We expect that the unified environmental context will allow us to describe in detail the interspecific differentiation, resulting from the speciation process. We hypothesise that sympatric populations will be affected by character displacement. We aim to discuss the observed patterns also in the perspective of environmental changes of the Anthropocene.

## 2. Materials and Methods

### 2.1. Specimens

Only adult specimens were used in our analyses, the age was estimated based on the date of death, dental abrasion [22], presence of milk teeth [20] and skull proportions [15,17]. A total of 69 skulls were examined (Appendix A), 29 of them represented *E. europaeus*, 25 *E. roumanicus* and 15 interspecific hybrids. The specimens originated from different localities (Figure 1) within the zone of sympatry (Czech Republic—14 *E. europaeus*, 13 *E. roumanicus*) and from allopatric localities in central Europe (Germany—15 *E. europaeus*, Slovakia—12 *E. roumanicus*). Samples from areas of sympatry were obtained from individuals that died in rescue centers. Skulls from areas of allopatry were borrowed from the Museum für Naturkunde, Berlin (Germany; MFN) and the Institute of Vertebrate Biology, Brno (Czech Republic; IVB). The skulls of interspecific hybrids deposited at MFN have also been examined. Seven such specimens originated from hybridisation experiments carried out in captivity by Herter [23], while eight hybrids were collected from the wild (w-hybrid) in Germany and the Czech Republic. Hybrid individuals show morphological traits of both species regarding classical morphometric indices as mentioned above [23].

### 2.2. Genetic Analyses

Considering the low frequency of interspecific hybridisation (one backcrossed individual out of 210 tested in Czechia and Slovakia) [3,4], species determination based on phenotypes has been confirmed by sequencing the mitochondrial control region in the samples originating from the zone of sympatry (N = 27). The tissue was collected in the course of skull preparation, and fixed in 96% ethanol, prior to storage at −20 °C. DNA was extracted using a DNA Blood and Tissue Kit (Qiagen, Prague, Czech Republic). Amplification of the mtDNA control region was achieved in line with the protocol used in Bolfíková and Hulva [3]. Sequences obtained were compared with those available in the GenBank^®^ database.

### 2.3. 3D Geometric Morphometrics Approach and Statistical Analyses

Skull sizes and shapes were assessed using three-dimensional landmark-based geometric morphometrics. A total of 47 three-dimensional coordinates were recorded corresponding to 13 landmarks on the mandible, and 34 landmarks on the cranium divided into 13 and 21 landmarks on the ventral and dorsal side, respectively (Figure 2, Appendix A). Landmark coordinates were acquired using the Reflex microscope and Axel software (Reflex Measurement Ltd., Butleigh, Somerset BA6 8SP, UK) at the Museum and Institute of Zoology of the Polish Academy of Sciences with headquarters in Warsaw. All measurements were taken by the same person (MK).

The coordinates were superimposed using a generalised Procrustes analysis algorithm [24,25]. During this procedure, all specimens are translated, so the center of gravity of their landmark configuration coincides, normalised to the unit centroid size and rotated to minimise the squared summed distances between corresponding landmarks. Centroid size (CS) corresponds to the square root of the sum of squared distances of the landmarks from their centroid [26]. The coordinates after superimposition correspond to the shape data.

Differences in the logarithm of CS between groups were depicted using a boxplot. Because of the small number of specimens in some of the comparisons, significance of the between-group differences was tested using the non-parametric Wilcoxon rank tests for two groups, and the Kruskal–Wallis tests when more than two groups were compared. Shape differences were tested using one-way multivariate analysis of variance (MANOVA) and canonical variate analyses (CVA) in combination with leave-one-out cross validation percentages (CVP), following Evin et al. [27], taking into account the unbalanced sample size between groups. Mean CVP values are provided with a 90% confidence interval obtained with 100 resamples [27]. Shape changes along the CVA axes were visualised by calculating shape changes along the factorial axes using multivariate regression [28] using the ‘Morpho v2.8′ R package [29]. Because sample sizes were relatively small compared to the large number of variables, we applied dimensionality reduction of the data prior to MANOVAs and CVAs by substitution of the primary data by the first scores of the principal component analyses (PCA) maximizing the leave-one-out cross validation between groups [27,30].

To test for possible common differences in size and shape between allopatric and sympatric populations of the two species, we applied two-way ANOVA and MANOVA using size or shape as variables, species as the main classifier, and allopatric versus sympatric distribution as a sub-classifier factor. When necessary, *p*-values were adjusted for multiple comparisons after Benjamini and Hochberg [31]. Overall phenotypic similarities between groups were depicted using neighbour-joining networks, computed based on the Mahalanobis *D*^2^ distances [32]. When allopatric and sympatric populations of the two species were compared, CVP were calculated for the main branches of the resulting neighbour-joining networks. The relationship between phenotypic data and geographic origin (latitude and longitude) was explored using the Mantel-test [33] for shape (Procrustes distances were used, and a randomized approach with 999 replicates), and regression [34] for centroid size for each structure and each species separately. To test whether the observed patterns of similarity can be explained by convergence [35], we adopted measures of the multidimensional convergence index (MCI) [36] calculated as the ratio of the Procrustes variance [37] within the putatively convergent lineages (i.e., sympatric populations) and within their sister lineages (i.e., allopatric populations). The obtained MCI values were compared to the distribution of 999 randomised MCI values (allopatric/sympatric attribution was randomised). Analyses were performed using the geomorph v3.2.0 [38] and ade4 v1.7.13 [39] for packages for R v3.6.3 [40].

## 3. Results

### 3.1. Differences between the Species

Morphologically related visual determinations of the species proved consistent with genetic assignment of animals within the sympatric population. On average, specimens of *E. europaeus* have a larger ventral side to the cranium and mandibles than those of *E. roumanicus* (Figure 3, Table 1), as well as a distinct cranial and mandibular shape (Table 1, Figure 4). When sympatric and allopatric populations are analysed jointly, 89.0% (CI: 86−92%), 94.8% (CI: 90−98%) and 84.0% (CI: 80−90%) of ventral, dorsal and mandibular shapes support the correct assignment of individuals to species (Table 1, Figure 4). If size alone is taken into account, the levels of correct cross-validation will drop respectively to 85.5% (CI: 84−88%), 78.2% (CI: 72−82%), and 72.7% (CI: 70−74%) (Table 1).

In terms of shape differences, *E. roumanicus* differs from *E. europaeus* in possessing a ventral size of the brain case that is proportionally more rounded (Appendix A), as well as a dorsal side of the skull that is proportionally narrower (Appendix A), and a mandible that is proportionally thinner, with the most anterior part shifted backward (Appendix A).

### 3.2. Contrasted Species Differentiation in Allopatry and Sympatry

When allopatric and sympatric populations are analysed separately, as opposed to via pooled analysis, the results are seen to differ. In allopatry, the species exhibit significant differences in sizes and shapes of crania and mandibles (Table 1), to the extent that the minimal mean cross-validation between them is of 84.2% for the size of the dorsal side of the cranium (Table 1). *Erinaceus roumanicus* shows a smaller skull size than *E. europaeus* in all comparisons (Figure 3).

In relation to both size and shape, mean cross-validation percentages for the sympatric populations are always lower than those obtained in allopatry (Table 1). The sympatric populations of *E. europaeus* and *E. roumanicus* only differ significantly in dorsal skull shape, as well as the size of the mandibular and ventral skull view, with *E. roumanicus* characterised by smaller ventral size (Figure 3).

### 3.3. Comparison of Allopatric and Sympatric Populations

Both species are of smaller size in sympatry (Figure 3), with a further effect being that the two are also more similar in size than they are in allopatry. CVA analyses thus reveal marked differences between allopatric and sympatric populations, as well as between the two species (Appendix A).

The overall phenotypic dissimilarity between the populations (Figure 5) shows a greater similarity between the sympatric populations of the two species than between the two (sympatric and allopatric) populations of the same species, when it comes to the ventral side of the cranium and the mandible (Figure 5a’,c’). In the case of these two structures, morphometric differentiation is more affected by allopatric/sympatric status than by taxonomy. Conversely, for the dorsal side of the cranium (Figure 5b’), the main differentiation is between the two species. The interaction term of the two-way analyses of variance reveals homogeneous patterns of differentiation between allopatric and sympatric populations of the two species (size and shape, all *p* > 0.5; Figure 5), with the one exception relating to mandible shape (*F*_14, 49_ = 3.27, *p* < 0.01).

MCI yielded values of 0.426 for the ventral and 0.441 for the dorsal side of the cranium, indicating the absence of convergence, while for the mandible, a convergence pattern with the MCI value of 1.541—well above one—and above 95% of the random MCI distribution (Figure 6), was detected.

### 3.4. Differences between the Two Hybrid Populations

Captive and wild hybrids do not differ in shape, in relation to any of the structures analysed (Table 2), though wild hybrids have a larger ventral side of the cranium than their captive counterparts (Figure 3). Neighbour-joining networks reveal closer proximity of the hybrids to the allopatric populations of the two species where data for the ventral side of the skull are concerned, as well as the mandible (Figure 5a,b). In relation to the dorsal side of the cranium, hybrids are seen to cluster between the two parent species (Figure 5c).

### 3.5. Geographical Structure

Unlike in the case of the dorsal side of the skull—for which no geographical structure could be detected, data for the ventral side and the mandible are seen to vary geographically (Table 3), with differences between the two species noted. The population of *E. europaeus* shows a geographical structure, as regards data on mandibular shape, while specimens of *E. roumanicus* also vary geographically in this respect, as well as in the size and shape of the ventral cranial side. However, the sampling on this remains limited and this conclusion is awaiting better support.

## 4. Discussion

### 4.1. Convergent Character Displacement in Size and Shape

Our first-ever reference to high-resolution three-dimensional geometric morphometrics in the case of hedgehogs helped reveal fine-scale differences between the two species in allopatry, while showing surprising similarity in circumstances of sympatry. Where matters of size are concerned, specimens of *E. europaeus* allopatric from *E. roumanicus* have larger crania and mandibles. In contrast, when present sympatrically, populations of the two species show a similar reduction in size, resulting in a much greater overlap than in allopatry, to the extent that cross-validation percentages are always lower in sympatric situations.

As regards shape, marked differences were detected for both species, between allopatric and sympatric populations. It emerged that most of the shape variation is clustered by allopatric/sympatric differences in ventral skull shape, as well as mandibles; while data for dorsal skull shape appear to be structured primarily by reference to inter-species differences. The overlap in the mandible variation, characterising sympatric populations in the discriminant analysis (Figure 4) confirmed by: their close proximity on the network (Figure 5); their lower percentage in cross-validation than their allopatric counterparts (Table 1); and the high MCI value, can be interpreted as a case of convergence in line with the pattern-based definition (Stayton 2015). However, interpretation of these results in accordance with the process-based definition of convergence (i.e., as bilateral convergent character-displacement) requires the examination of possible microevolutionary factors responsible for the presence of more-unified phenotypic traits under circumstances of sympatry.

### 4.2. Hybridisation and Introgression

Hybridisation might be the simplest explanation for the observed patterns, with hybrids in this case known to be large, and more similar in size to *E. europaeus* (perhaps as a reflection of the genome dominance of this species when it comes to the determination of size; or else in line with asymmetry in the ability to produce back-crosses). The experiments of Poduschka and Poduschka [41], combined with our data, suggest that asymmetrical introgression is to be anticipated (and therefore also asymmetrical changes in morphology).

In terms of shape, hybrids fall between the two parental allopatric morphologies when it comes to mandibles, and the ventral side of the skull. This is consistent with the renewal of the plesiomorphic state in hybrids. Interestingly, as captive and wild hybrids do not differ in skull shape (while the former have larger crania and mandibles than their wild counterparts), the suggestion is that populations held in isolation may develop different morphological traits [42,43].

Although past introgression events may have had a role to play in hedgehogs [4], the recent level of hybridisation is very low [3,5]. However, a genomic approach and ascertainment of the level of ancient introgression will be necessary if the potential role of hybridisation in phenotypic evolution is to be investigated fully.

### 4.3. Clinal Variation

A second potential explanation would involve general trends to the clinal variation characteristics of both species, possibly affected by the same selective environmental gradient that overrides the effect of competition [44]. However, given the limited width of the contact zone between the hedgehog species, it is difficult to imagine a longitudinal gradient taking on extreme values in central Europe. For example, Škoudlín [45] measured 23 metric characters and four proportional indices related to the skull, in order to compare specimens of *E. roumanicus* from the Czech Republic, Poland and Belarus. Those from Poland and Belarus were characterised by higher values for a majority of the characters studied, but the latitudinal pattern was rather in line with Bergmann’s rule. A longitudinal cline has never been referred to in hedgehogs.

### 4.4. Ecological Species Interactions—Competitive and Predator Pressure

A third hypothesis would involve biotic ecological interactions, i.e., competitive pressure, as a possible causal factor accounting for patterns observed. The general expectation of divergent character displacement is based on a presumption of sufficiently narrow ecological valence, with resource competition generating selective pressure intense enough to promote niche diversification. It is well known that much of the diversity among terrestrial vertebrate skulls is associated with feeding [6,46,47] and that the adaptive evolution associated with it can be rapid [48,49,50]. Hedgehogs are medium-sized plantigrade mammals whose low basal metabolic rates and related defensive anti-predator strategies reflect energetic constraints associated with the unpredictable distribution of resources and predation [51]. Therefore, when two hedgehog species form a sympatric zone, potentially characterised by higher population density than in allopatry, the resulting guild may approach the carrying capacity of the environment, with competition for resources generating a convergent reduction in body size, rather than divergent character displacement. This explanation is also consistent with convergence in the mandible shape. As the function of the lower jaw connects mainly with feeding [47,52], in contrast to the cranium reflecting additional functions related to sensory organs and the brain [53,54], it is reasonable to ascribe respective patterns to evolutionary forces within the trophic niche. We hypothesize that decrease in size and convergence in shape is a result of local adaptations to overlapping dietary resources in the two species, and hence an example of convergent character displacement.

Background mechanisms underpinning the observed patterns could also act at the community level and entail predator pressure. The disruptive selection this causes could be seen as a proximate mechanism behind a bimodal distribution of body size in mammals, in line with Cope’s rule [55]. Mammals filling “the mid-size gap” often possess defensive weaponry apomorphies, such as the spines in hedgehogs [56], thereby pointing to the major evolutionary consequences of predation among such medium-sized mammals. Recent ecological research on hedgehogs also reported a marked effect of intraguild predation [57]. The associated hypothesis accounting for our results would therefore involve higher predator pressure in the sympatric zone, as a reflection of the higher population density among hedgehogs in general, and the recent mesopredator release [58], i.e., population increases in e.g., red foxes and European badgers in central Europe. That may in turn reflect the population declines that characterised large carnivores for a long period in the past. All of this leaves evolution in the direction of smaller body size in prey species as more suitable for achieving crypsis. These ecological mechanisms could all be facilitated by such anthropogenic changes in the environment as declines in numbers of invertebrates due to the use of pesticides, fragmentation of habitats, or increased traffic.

As detailed knowledge about ecological niche of hedgehogs is crucial in understanding their population decline [59] and proposing rational conservation management, we propound further attention to described phenomena. Considering the substantial regional variation and steep east–west gradients in socioeconomic variables in Europe, resulting in geographic patterns in landscape fragmentation [60] and anthropogenic pressure on ecosystems and hedgehog populations, as well as differences in conservation management in rescue centers [61], large-scale studies using a standardized methodology are needed to compensate for possible observation bias and investigate more closely the complex phenotypic variation patterns and ecological niche characteristics. Taking into account environmental variables will also allow for the study of the presence of complex phenomena caused by selection pressures related to urbanization, the development of industrialised agriculture and other human-mediated factors such as anthropogenic dwarfing of species.

## 5. Conclusions

In the present study, seldom observed sympatric character displacement was ascertained in the contact zone of two sympatric hedgehog species in central Europe based on the analysis of skull size and shape. Considering almost complete formation of reproductive isolating barriers, we presume hybridisation and introgression are not the major processes behind the observed pattern. We hypothesise that unidirectional selection to size and shape related to competition in the trophic niche is responsible for that pattern, and that size changes of the skull could also be facilitated by a reduction in body size related to selection pressures, mediated by the Anthropocene mesopredator release.

## Figures and Tables

**Figure 1 animals-10-01803-f001:**
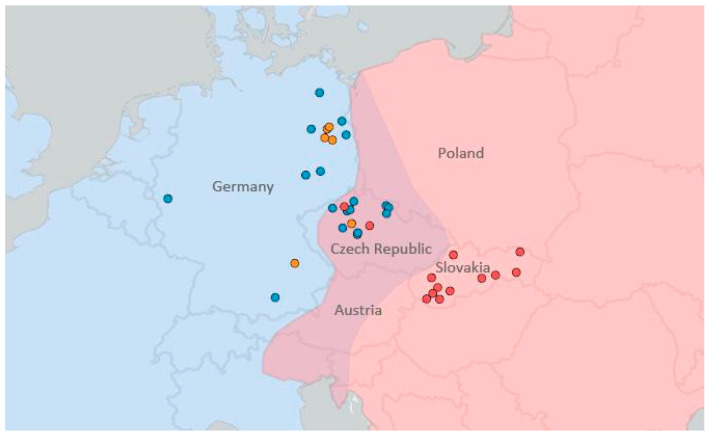
Species ranges and localities of samples used in the present study. The species are marked in different colours: blue represents *E. europaeus*, red represents *E. roumanicus* and orange indicates wild hybrids of the two species. The contact zone position (based on available data) is highlighted in violet. The map was created using ArcGIS Online.

**Figure 2 animals-10-01803-f002:**
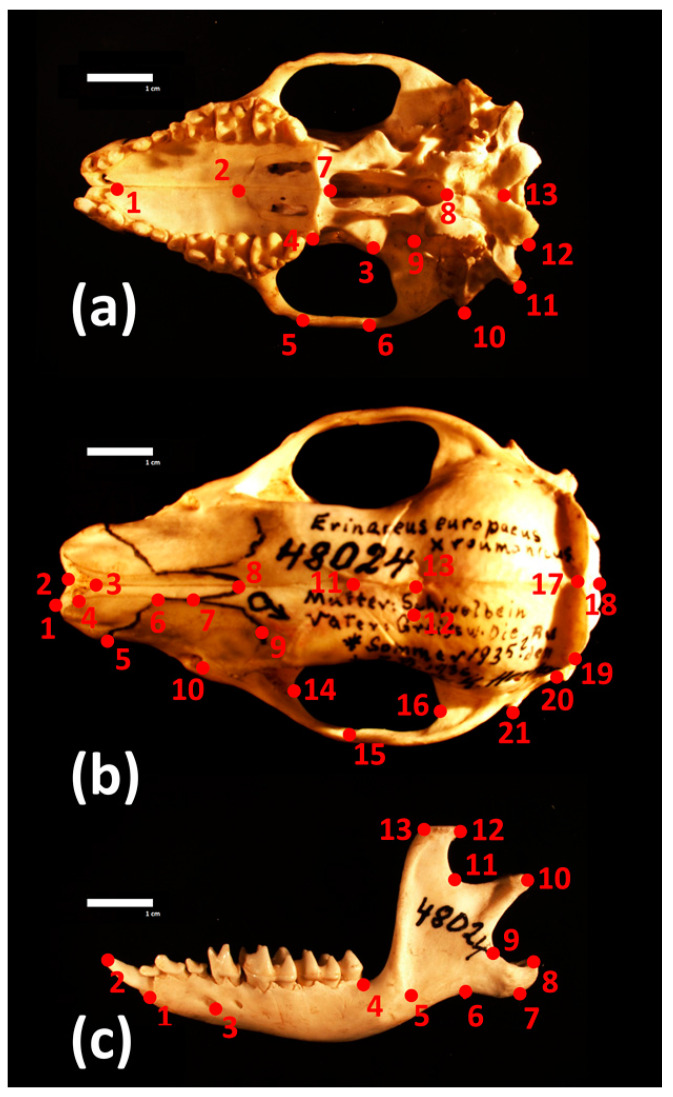
Location of the three-dimensional landmarks measured on (**a**) ventral and (**b**) dorsal sides of the cranium, and (**c**) the mandible.

**Figure 3 animals-10-01803-f003:**
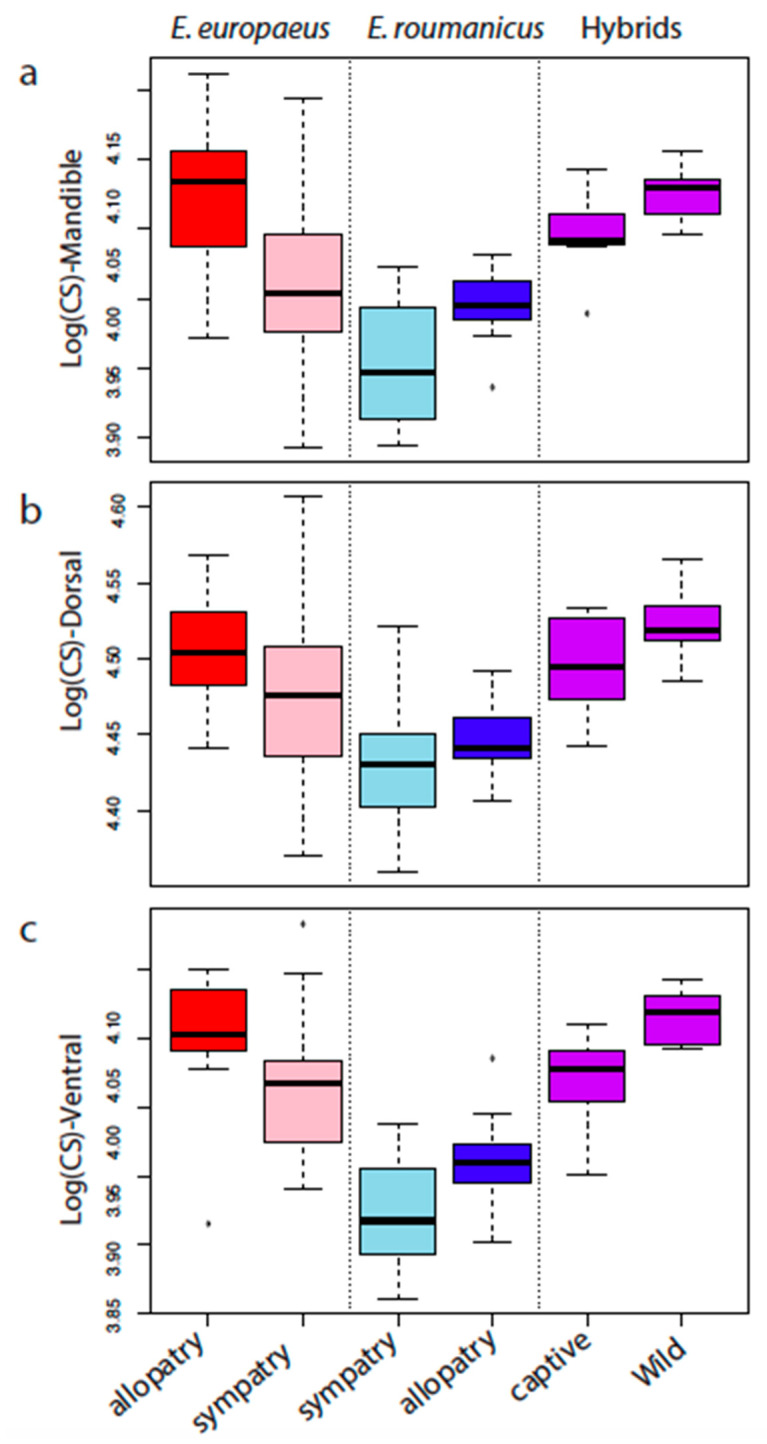
Cranial and mandibular size differences and variability between allopatric, sympatric and hybrid (captive and wild) populations of *E. europaeus* and *E. roumanicus*. Boxplots of the log centroid sizes (CS) of the mandible (**a**), dorsal (**b**) and ventral (**c**) sides of the cranium.

**Figure 4 animals-10-01803-f004:**
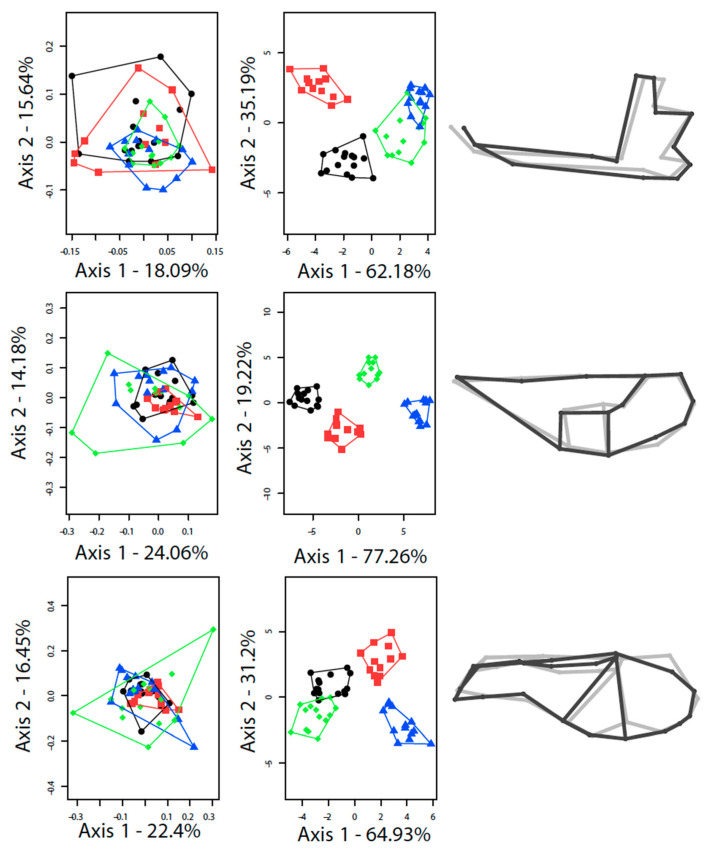
Principal component analysis (PCA; left section), discriminant analysis (DA; central section) and visualisation of the differences in shape between the two species (right section) in relation to mandible (top), ventral side (middle) and dorsal side (bottom). For PCA and DA, black stands for *E*. *europaeus* in allopatry, green for *E. europaeus* in sympatry, red for *E. roumanicus* in allopatry, blue for *E. roumanicus* in sympatry. For shape comparison, black stands for *E. roumanicus* and grey for *E. europaeus*; shape differences are amplified by a factor of 5.

**Figure 5 animals-10-01803-f005:**
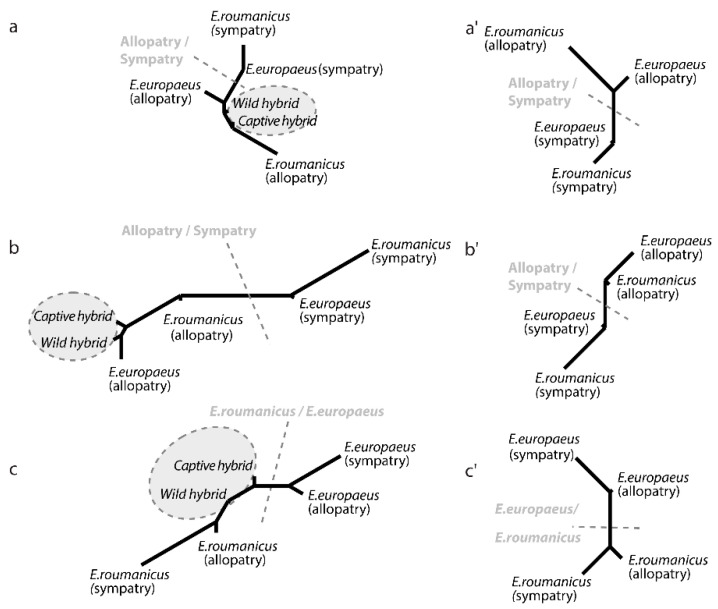
Overall shape differences among examined groups of *E. europaeus* and *E. roumanicus*: Right—between sympatric and allopatric populations of the two species; left—among all groups (allopatric, sympatric and hybrid populations). Neighbour-joining networks of the Mahalanobis distances for the mandible (**a**,**a’**), and the ventral (**b**,**b’**) and dorsal (**c**,**c’**) sides of the cranium.

**Figure 6 animals-10-01803-f006:**
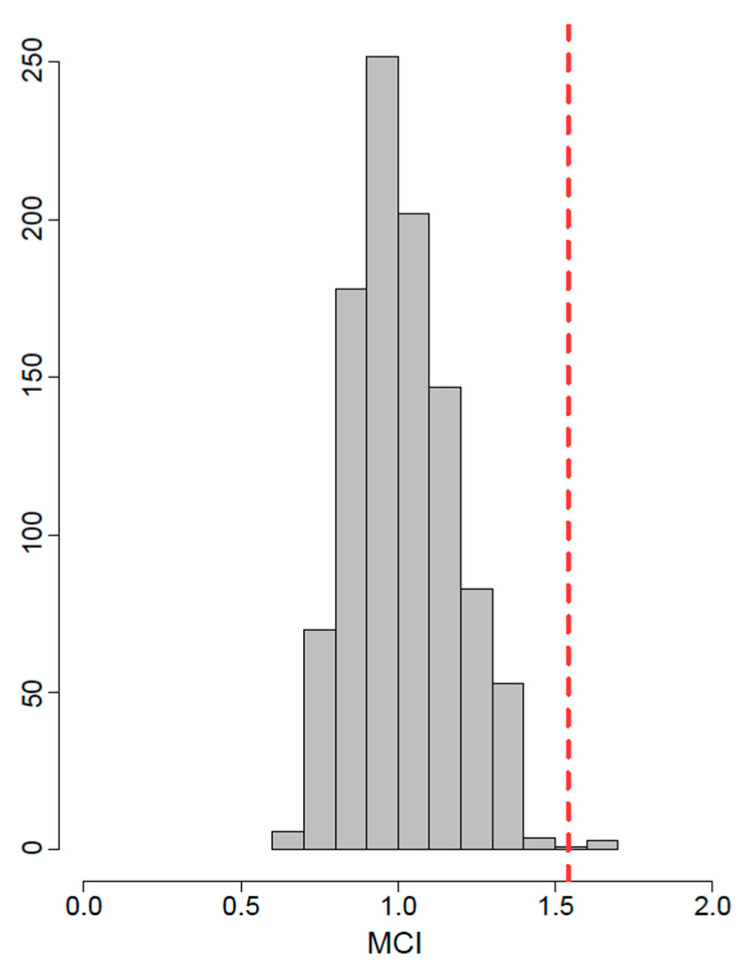
Multidimensional convergence index (MCI) of the sympatric and allopatric populations of both species obtained for the mandible (vertical dotted line) compared with a randomized distribution of MCI values.

**Table 1 animals-10-01803-t001:** Test of differences between species: overall (pooling sympatric and allopatric specimens) and separately sympatric or allopatric populations only. Differences in shape and size were tested using one-way multivariate analysis of variance (MANOVA) and Wilcoxon’s tests, respectively, for ventral and dorsal sides of the cranium and the mandible. P: *p*-value, numDf and denDf: Numerator and denominator degrees of freedom, W: statistic of the Wilcoxon’s tests. The numerator degree of freedom corresponds to the number of Principal Component (PC) scores included in the analyses. Cross-validation percentages (CVP) are provided as the mean and the 90% confidence interval of the distribution.

Trait		Shape			Size		
		F(numDf, denDf)	*p*	CVP	W	*p*	CVP
Mandible	Overall	F(22,30) = 5.94	5 × 10^−6^	83.3% (80–88.1%)	16.648	4 × 10^−5^	72.5% (70–76%)
	Sympatry	F(15,10) = 2.16	0.11	-	134	0.01	69.2% (69.2–69.2%)
	Allopatry	F(13,13) = 11.4	4 × 10^−5^	89.8% (79–95.8%)	164	1 × 10^−4^	85.7% (79–87.7%)
Ventral	Overall	F(23,27) = 8.68	2 × 10^−7^	88.7% (84–92.1%)	25.12	5 × 10^−7^	85.3% (84–88%)
	Sympatry	F(19,5) = 4.47	0.05	-	133	0.04	66.6% (61.5–69.2%)
	Allopatry	F(10,15) = 7.32	3 × 10^−4^	92.3% (91.7–95.8%)	155	7 × 10^−5^	92.2% (91.7–95.8%)
Dorsal	Overall	F(29,24) = 12.41	1 × 10^−8^	95.0% (92–98%)	15.713	7 × 10^−5^	78.2% (72–82%)
	Sympatry	F(19,7) = 17	4 × 10^−4^	90.8% (84.6–100%)	145	7 × 10^−5^	79.5% (79.2–83.3%)
	Allopatry	F(15,11) = 6.6	1 × 10^−3^	84.25% (83.3–87.5%)	167	4 × 10^−5^	84.7% (83.3–87.5%)

**Table 2 animals-10-01803-t002:** Test of differences between captive and wild hybrids. Differences in shape and size were tested using MANOVA and Wilcoxon’s tests, respectively, for the ventral and dorsal sides of the cranium and the mandible. P: *p*-value, numDf and denDf: Numerator and denominator degree of freedom. Cross-validation percentages (CVP) are provided as the mean and the 90% confidence interval (CI) of the distribution.

Trait	Mandible		Ventral		Dorsal	
	Tests	CVP	Tests	CVP	Tests	CVP
Shape	F(2,12) = 0.6, *p* = 0.56	-	F(2, 12) = 2.8, *p* = 0.09	-	F(10,4) = 4.08, *p* = 0.09	-
Size	W = 46, *p* = 0.04	73.9% (64.3–85.7%)	W = 52, *p* = 0.004	79.4% (71.4–92.9%)	W = 40, *p* = 0.19	-

**Table 3 animals-10-01803-t003:** Geographic structure of the data. Results of Mantel tests for shape and regression for size for ventral and dorsal sides of the cranium and the mandible.

Trait	*E. europaeus*				*E. roumanicus*			
	obs	*p*-Value	adj. R2	*p*-value	obs	*p*-Value	adj. R2	*p*-Value
Mandible	0.283	0.036	0.167	0.039	0.221	0.005	0.274	0.011
Ventral	−0.09	0.775	0.04	0.232	−0.066	0.762	0.18	0.042
Dorsal	−0.070	0.626	−0.01	0.44	−0.014	0.53	−0.02	0.47

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
