# Peer review of "3D Geometric Morphometrics Reveals Convergent Character Displacement in the Central European Contact Zone between Two Species of Hedgehogs (Genus Erinaceus)"

_animals, 2020, doi:10.3390/ani10101803_

Round 1

Reviewer 1 Report

This is an interesting study and the authors have collected a unique dataset. The paper is generally well written and structured.

Author Response

Thank you.

Reviewer 2 Report

I appreciate the effort made by the authors to improve their manuscript, as well as the responses and clarifications provided by them in their response letter. Nevertheless, there are still some points that need to be addressed prior to an eventual publication.

General comments:

  1. Convergence: As I mentioned in my previous review, the authors consider results that their observed results are due to convergent directional selection towards smaller skull size, as well as convergence in shape as a result of local adaptations for overlapping dietary resources. However, the authors do not provide a definition of convergence, which is certainly fundamental to understand why and how they will test whether the observed morphological patterns are the product of evolutionary convergence or not. The authors correctly point out that they cannot test convergence using the methods described in the references I previously provided (as this manuscript focuses on smalls samples with a limited phylogenetic extent). However, these papers were suggested as general reference materials that show that studying convergence is not as simple as merely finding some phenetic similarities, but rather depends on how convergence is defined and measured (aspects that are still lacking in the manuscript). As mentioned by Stayton (2015a, 2015b) one of the main problems when comparing studies focused on convergent evolution results from the fact that inferences about convergence are often compromised by inconsistencies in the definitions, measures, significance tests and inferred causes of convergent evolution. For example, some researchers define convergence as a mere evolutionary pattern (e.g. Starr et al. 2015), while others consider developmental and/or adaptive factors (e.g. Pagel 2002). Relatedly, some researchers prefer to quantify convergence using a geometric approach (e.g. Stayton 2006), whereas others quantify it relative to adaptive peaks by assuming certain underlying evolutionary mechanisms (Ingram & Mahler, 2013). Hence, it is fundamental that the authors explicitly explain in the text how they define convergence and based on that, decide what will be the most appropriate test to be used. Only then they would be able to properly discuss whether convergence is the evolutionary factor underlying the observed phenotypic trends. The fact that the authors seem to equate mere morphometric similarity with convergence, makes me wonder if they understand what the concept of convergent evolution actually entails.

  • Stayton, C. T. (2015a). The definition, recognition, and interpretation of convergent evolution, and two new measures for quantifying and assessing the significance of convergence. Evolution, 69(8), 2140–2153. https://doi.org/10.1111/evo.12729
  • Stayton, C. T. (2015b). What does convergent evolution mean? The interpretation of convergence and its implications in the search for limits to evolution. Interface Focus, 5(6), 20150039. https://doi.org/10.1098/rsfs.2015.0039
  • Starr C, Evers CA, Starr L. (2015). Biology: concepts and applications, 9th edn. Stamford, CT: Cengage Learning.
  • Pagel M. (2002). Encyclopedia of evolution, 1. Oxford, UK: Oxford University Press.
  • Ingram, T., & Mahler, D. L. (2013). SURFACE: Detecting convergent evolution from comparative data by fitting Ornstein-Uhlenbeck models with stepwise Akaike Information Criterion. Methods in Ecology and Evolution, 4(5), 416–425. https://doi.org/10.1111/2041-210X.12034
  • Stayton CT. (2006). Testing hypotheses of convergence with multivariate data: morphological and functional convergence among herbivorous lizards. Evolution 60, 824–841

  1. Geometric morphometrics (GM): As I also explained in my previous review, many of the statistical procedures that were carried out are not well justified and do not correspond to standard GM practice. This did not necessarily mean that they were wrong, but rather that they should be better justified and explained in the text, as any researcher interested in the procedures that you applied would appreciate additional details. At no point did I put into question the expertise or credentials of any of the authors. As such, I consider that it is inappropriate to provide a link to a Google Scholar profile, since ‘appealing to authority’ does not resolve in any way the points I previously raised. In fact, many of the problems I previously described are still there:

Intra-observer error is still unknown: The authors did not measure intra-observer error, which is of fundamental importance in GM. As clearly shown by Fruciano (2016), measurement error is essential in empirical scientific work and morphometrics is no exception. However, it is surprisingly a factor that is frequently disregarded or not even considered in many GM empirical studies (which certainly does not justify not measuring this source of error). This is potentially an extremely serious matter since random measurement error can inflate the amount of variance, which can result in loss of statistical power. Hence, the explanation provided by the authors that the specimens “were loaned from the museums for a limited amount of time and we have no possibility to check it again” is certainly insufficient and not satisfactory. There are several possible solutions that they could have tried to measure their error in spite of the fact the original sample is no longer available. For instance, they could measure twice (or more times) a limited number of new specimens of the same species and then carry out a Procrustes ANOVA (Klingenberg, & McIntyre, 1998) to estimate your error. This is not an unreasonable request, on the contrary, measurement error should be always included as part of GM analyses (Fruciano, 2016).

  • Fruciano, C. (2016). Measurement error in geometric morphometrics. Development Genes and Evolution, 226(3), 139–158. https://doi.org/10.1007/s00427-016-0537-4
  • Klingenberg, C. P., & McIntyre, G. S. (1998). Geometric Morphometrics of Developmental Instability: Analyzing Patterns of Fluctuating Asymmetry with Procrustes Methods. Evolution, 52(5), 1363–1375. https://doi.org/10.2307/2411306

Additional GM details:

 In my previous review, I mentioned that the authors should explain why they carried out this multivariate regression, as it is possible to visualize shape differences along the CVA axes using the obtained CV scores in a direct fashion. The authors replied that “Multivariate regression is a way to calculate shape visualization along CVA axis, but we agree not the most common one, though published in high impact journals”. However, they do not provide references showing that multivariate regression is a way to calculate shape visualization along CVA axes (please include these references). The only reference provided corresponds to Monteiro (1999), which corresponds to an excellent article on Multivariate Regression Models in GM. However, nowhere in that article, there is any reference to CVA or why would anyone visualize the results from a CVA using a regression (which is not wrong but should be properly explained in your manuscript). Please briefly explain the rationale underlying your procedure and provide references.

  • Monteiro, L. R. Multivariate Regression Models and Geometric Morphometrics: The 445 Search for Causal Factors in the Analysis of Shape. Syst. Biol. 1999, 48 (1), 192–199. 446 doi:10.1080/106351599260526.

They also mentioned that “The review does not provide information about the way he/she would had like to have it done”.  Shape changes associated with the canonical variates (CVs) can be visualized using Schlager’s excellent ‘Morpho’ package (since you are using R) https://www.rdocumentation.org/packages/Morpho/versions/2.8/topics/CVA

  • Schlager S (2017). “Morpho and Rvcg - Shape Analysis in R.” In Zheng G, Li S, Szekely G (eds.), Statistical Shape and Deformation Analysis_, 217-256. Academic Press. ISBN 9780128104934.

RV test: The authors state that “The RV has been demonstrated to be more suitable for geometric morphometric data than the traditionally used Mantel’s test”, without providing any reference(s) to support this claim (i.e., that the RV coefficient is more appropriate that Mantel test in the context of GM). Although it is true that Mantel’s test has been criticized (e.g., especially when data is autocorrelated; Guillot & Rousset, 2013), the RV coefficient has been particularly criticized in the context of GM (Adams, 2016; Adams & Collyer, 2016; Bookstein 2016), and its use has been not recommended when using landmark data. Even though the RV coefficient might be an intuitive way of measuring the correlation between blocks of data. (i.e., it corresponds to a multivariate generalization of the squared correlation coefficient), it possesses mathematical deficiencies that are particularly unsuitable when dealing with GM data. The RV coefficient is sensitive to both the sample size of the data set and the number of variables under analysis (Adams, 2016). Consequently, when applying the RV coefficient, covariation patterns present in data may be confounded with trends generated by the number of variables, as well as sample size. In fact, this precisely the reason why the RV coefficient was removed from ‘geomorph’, which is one of the R packages used by the authors. Bookstein (2016) even claims that “From the proper understanding of the RV coefficient it will follow, […], that it is valueless in most organismal applications”. Hence, I honestly consider that a better option would be to apply a Mantel (or partial Mantel depending on the specific design) test stating the known criticisms and limitations of this method, as well as including relevant references. Alternatively, the authors could apply other association metrics and justify their use. For example, Josse & Holmes (2016) recommend that if measures such as the RV coefficients are being used, both their linear and nonlinear versions, as well as their bias-corrected versions should be used, reported, and compared.

  • Guillot, G., & Rousset, F. (2013). Dismantling the Mantel tests. Methods in Ecology and Evolution, 4(4), 336–344. https://doi.org/10.1111/2041-210x.12018
  • Adams, D.C. 2016.Evaluating modularity in morphometric data: Challenges with the RV coefficient and a new test measure. Methods in Ecology and Evolution 7:565-572.
  • Adams, D. C., & Collyer, M. L. (2016). On the comparison of the strength of morphological integration across morphometric datasets. Evolution, 70(11), 2623–2631. https://doi.org/10.1111/evo.13045
  • Bookstein, F. L. (2016). The Inappropriate Symmetries of Multivariate Statistical Analysis in Geometric Morphometrics. Evolutionary Biology, 43(3), 277–313. https://doi.org/10.1007/s11692-016-9382-7
  • Josse, J., & Holmes, S. (2016). Measuring multivariate association and beyond. Statistics Surveys, 10, 132–167. https://doi.org/10.1214/16-SS116

Minor points:

  • Please include the versions R and packages you are using in the main text.
  • Please slightly increase the scale factor/ magnitude in your shape change visualizations to better visualize shape changes associated with each one of your axes.
  • Please include in the text how many PCs (i.e., specific number) did you use in your CVAs and MANOVAs. This information is still missing in the text.

Round 2

Reviewer 2 Report

I appreciate the effort made by the authors to modify their manuscript, as well as the answers and clarifications provided by them in their response letter. I consider that the manuscript has been significantly improved and I am satisfied by the authors' responses. Hence, I can recommend this article for publication in Animals.

This manuscript is a resubmission of an earlier submission. The following is a list of the peer review reports and author responses from that submission.

Round 1

Reviewer 1 Report

Bolfíková et al uses 3D morphometrics to find traits that might be diagnostic to differentiate the hedgehog species E. europaeus and E. roumanicus. They find that, the shape of the dorsal side of the cranium is the best diagnostic trait. For the mandible, and ventral side of the cranium the populations form both species in the contact zone converge. Moreover, individuals from these populations are smaller in size. The authors propose that competition due to high density of animals in the contact zone and higher predator pressure may result in a reduction of size.

The manuscript is interesting, well structure, the methodology is overall adequate, and the conclusions seem plausible. Thus, I only have a few minor comments.

Introduction

Line 56: „Recent gene flow between E. europaeus and E. roumanicus has not been intensive.”. Describing the geneflow between both species as “sparse” seams more precise than “not intensive”.

Lines 57 to 63: The authors use the term “nascent” several times as “nascent niche”, “nascent habitat” and “nascent species”. It is not clear the meaning of nascent here. Does it mean newly formed? Since it is not a commonly used term, please add a sentence clarifying its meaning this for readers that might not be familiarized with it.

Material and methods

Genetic analysis: Although I understand it is unlikely to find hybrids, the authors should have used a method able to detect them molecularly. Since they are looking for diagnostic traits to differentiate the species, it is important to exclude the possibility of cryptic hybridization. I understand it might not be worthy to produce new data, so I suggest the authors to provide statistics based on previous work of the number of hybrids and backcrosses found in the contact zone. This way the readers will have a better idea of how unlikely of including undetected hybrids.

Results

Lines 183 to 186: The authors site Fig. S3, however there is no Fig. S1 and Fig. S2. I suggest changing the name of Fig. S3 to Fig. S1.

Discussion

The main explanations given by the authors (competition and predation) explain the reduction in size but not the converging shapes. The authors should also include an explanation for that. The authors already hinted that hedgehogs may explore the same trophic niche in sympatry which may have led to the reduction of size. The existence of the same food resources may have contributed to a convergence in feeding related traits through local adaptation. The authors show that, for example mandible, shows geographic structure which could be related to local adaptation.

Tables and pictures

Figure 1. Change the color of the landmarks. They are hard to see.

Figure 2. Better resolution

Figure 3: Figure needs better resolution. The legend says that the pictures in the right panel represent analyses with hybrids, however these are in the left panel. Also, it mentions that cross validation values are shown as percentage, but no value is present in the pictures.

Tables 1 and 2. The legend says that not significant results are highlighted in gray. That is not the case.

Table 3: What is the second table and what do N and E mean?  Also contrary to what the legend say non-significant values are not highlighted in gray. Nevertheless, significant values are in bold but this is not mentioned in the legend.

Table S1. I think it would be informative to add if the individual comes from the sympatric or allopatric population.

Author Response

Thank you for the valuable comments, see our comments in the attached document.

Reviewer 2 Report

  • I would suggest to include the taxonomic genus into the title (and remove it from Abstract)
  • Taxonomic names must appear in italics (some paragraphs, specially preliminary ones, lack it)
  • Line 56 is not clear
  • Perhaps a distribution map on Introduction section would help to understand this sympatric area
  • Line 130: no replica?
  • Line 134 must not appear "alone"
  • Line 140: use CS
  • line 143: clarify MANOVA ("Multivariate Analysis of Variance"), as you did for CVA
  • Figure 1 is not very clear. I suggest to increase size of landmarks dots.

The authors should be congratulated for conducting an important study.

Author Response

Thank you for the valuable comments, see our comments in attached document.

Reviewer 3 Report

This research compares the skull morphology of allopatric, sympatric, and hybrid hedgehogs by means of geometric morphometrics. In order to do so, the authors analyzed a total of 69 adult skulls using 3D geometric morphometrics and a battery of different statistical tests. The specific objectives of this work are 1) to compare shape and size differences between allopatric and sympatric E roumanicus and E. europaeus (both at the intra- and inter-specific levels); 2) to assess whether there is geographical structuration of the data, and 3) to interpret the observed patterns in an eco-evolutionary framework. Although this manuscript certainly focuses on an interesting topic, I consider that the work requires substantial work prior to an eventual publication in any journal, as it will become clear by reading the comments provided below:

General comments:

- English editing is needed as there are many sections that are not clear.

-There is an absence of clear and explicit hypotheses in the introduction.

-There are several sentences in the manuscript that should be referenced and there are many relevant citations missing (see annotated pdf)

-The figures are pixelated and in many cases not really clear. The overall quality of the figures is poor and they should be improved. (see annotated pdf for additional comments).

-In addition, the supplementary figures were not provided (the supplementary file only contains an excel table).

- Part of the discussion focuses on convergence but the authors did not explicitly test for convergence. Hence, they could try to either test if their data shows convergence or, alternatively modify their discussion, as in its current form relating their results to convergence is speculative.

-Finally, many of the statistical procedures that were carried out are not well justified and do not correspond to standard geometric morphometric practice (which means that should be better justified and explained). I provide further information about this below:

Methods:

  • The procedure used to determine age should be better explained and incorporated as a supplementary material (e.g. how were skull proportions used, or dental abrasion, etc.)
  • The Genetic analyses section should be expanded as it is extremely brief (i.e. how the DNA sequences were compared? did you run any analyses using the genetic data? did you compute any metrics using this data? , how did you align the sequences? etc.
  • Intra-observer error was not measured. You could, for example, digitize a sub-sample and carry out a Procrustes ANOVA to assess this potential source of measurement error.
  • There are several key morphometrics references missing in the section explaining the basic geometric morphometric procedures (lines 134-139).
  • The authors do not explain why they used non-parametric statistics when testing for size differences. Please explain and/or include normality tests.
  • Although the other correctly corrected significance (due to the multiple comparisons problems) the sample size of the study seems too limited for the number of comparisons the authors are performing. A power test could be used to assess the sample size required to properly address this issue.
  • (lines 146-147 ): The authors state that: differences along the CVA axes were visualized by calculating shape changes along the factorial axes using multivariate regression [28]. The authors should explain why they carried out this multivariate regression, as it is possible to visualize shape differences along the CVA axes using the obtained CV scores (there is no need to perform a regression to this). In addition, no shape visualizations are provided as part of the manuscript, which is intriguing, considering that one of the main advantages of geometric morphometrics, as compared to traditional linear morphometrics, is the possibility of easily visualizing shape changes (this could be in part due to the fact that some supplementary files seem to be missing).
  • The authors carried out a PCA but they do not provide that figure, which should be certainly incorporated as part of the manuscript (as well as providing the associated shape differences with these axes).
  • Additionally, the authors used the PCA as a dimensionality reduction tool before carrying out the CVAs and MANOVAs, but it is not clear how many PCs did they consider? how did they decide the numbers of PCs to be considered? this should be explained. 
  • It is unclear why the authors used the RV test as a way to test for geographical correlation. This test has been heavily criticized as this coefficient is adversely affected by data attributes (i.e., sample size and the number of variables), which means that it does not characterize the covariance structure between sets of variables in an appropriate way. The authors should at least justified their decision.  
  • Morphological descriptions of the shape changes along the main PCA and CVA axes should be provided, as well as figures showing these shapes.
  • The authors focused part of the discussion on convergence. However, no convergence test and/or metric was used. There are several approaches to test for convergence. If interested in this possible factor they should explicitly test for convergence in order to properly discuss this. See for e.g the following papers for some different approaches to analyze convergence: Ingram T, Mahler DL. SURFACE: detecting convergent evolution from comparative data by fitting Ornstein-Uhlenbeck models with stepwise Akaike information criterion. Methods Ecol Evol. 2013;4:416–25.
    Stayton CT. Is convergence surprising? An examination of the frequency of convergence in simulated datasets. J Theor Biol. 2008;252:1–14.
    Stayton CT. The definition, recognition, and interpretation of convergent evolution, and two new measures for quantifying and assessing the significance of convergence. Evolution (N Y). 2015;69:2140–53.
    Arbuckle K, Minter A. Windex: analyzing convergent evolution using the wheatsheaf index in R. Evol Bioinforma. 2015;11:11–4.

Additional comments:

-Several additional comments are provided in the annotated pdf.

-Check that all species names are in italics.

- Tables I and III say 'Mandibula' it should be 'Mandible'

Author Response

Thank you for the comments, see our comments in the attached document.
